# Effect of Evening Primrose Oil Supplementation on Selected Parameters of Skin Condition in a Group of Patients Treated with Isotretinoin—A Randomized Double-Blind Trial

**DOI:** 10.3390/nu14142980

**Published:** 2022-07-21

**Authors:** Agnieszka Kaźmierska, Izabela Bolesławska, Adriana Polańska, Aleksandra Dańczak-Pazdrowska, Paweł Jagielski, Sławomira Drzymała-Czyż, Zygmunt Adamski, Juliusz Przysławski

**Affiliations:** 1Department of Bromatology, Poznan University of Medical Sciences, 60-806 Poznan, Poland; a.kazmierska@interia.pl (A.K.); drzymala@ump.edu.pl (S.D.-C.); jprzysla@ump.edu.pl (J.P.); 2Laboratory of Connective Tissue Diseases, Department of Dermatology and Venereology, Poznan University of Medical Sciences, 60-355 Poznan, Poland; apolanska@ump.edu.pl; 3Department of Dermatology, Poznan University of Medical Sciences, 60-806 Poznan, Poland; aleksandra.danczak-pazdrowska@ump.edu.pl (A.D.-P.); zadamski@ump.edu.pl (Z.A.); 4Department of Nutrition and Drug Research, Institute of Public Health, Faculty of Health Sciences, Jagiellonian University Medical College, 31-066 Krakow, Poland; paweljan.jagielski@uj.edu.pl

**Keywords:** acne, isotretinoin, evening primrose oil, skin hydration, transepidermal water loss, sebum, body weight, Body Mass Index

## Abstract

Background: Retinoids, which include isotretinoin, reduce sebum levels, the degree of epidermal wetness (CORN) and cause an increase in transepidermal water loss (TEWL). Weight gain has also been observed in isotretinoin-treated patients. An agent that can reduce the severity of isotretinoin side effects is evening primrose oil (*Oenothera paradoxa*). The purpose of this study was to evaluate the effect of evening primrose oil supplementation in patients with acne vulgaris treated with isotretinoin on skin hydration status (CORN), transepidermal water loss (TEWL), skin oiliness (sebum) and changes in body weight and BMI. Methods: Patients diagnosed with acne were assigned to the isotretinoin-treated group (*n* = 25) or the isotretinoin and evening primrose oil-treated group (*n* = 25). The intervention lasted 9 months. CORN (with a corneometer), TEWL (with a tewameter) and sebum (with a sebumeter) were assessed twice, as well as body weight and BMI (Tanita MC-780). Results: The isotretinoin-treated group showed statistically significant reductions in CORN (*p* = 0.015), TEWL (*p* = 0.004) and sebum (*p* < 0.001) after the intervention. In the group treated with isotretinoin and evening primrose oil, TEWL and sebum levels also decreased significantly (*p* < 0.05), while CORN levels increased from 42.0 ± 9.70 to 50.9 ± 10.4 (*p* = 0.017). A significant decrease in body weight (*p* < 0.001) and BMI (*p* < 0.001) was observed in both groups after 9 months of intervention. Conclusions: During isotretinoin treatment, supplementation with evening primrose oil increased skin hydration. However, there were no differences between groups in transepidermal water loss, skin oiliness, weight loss and BMI.

## 1. Introduction

Acne vulgaris, a chronic inflammatory disease of the outlets of the hair follicles and sebaceous glands, occurs in 85% of all young people aged 12–24 and in 3–43% of adults [1,2]. In about 20 to 25% of individuals, there is progression of acne to a severe form [3], causing patients to develop permanent scarring [2], social isolation [4], lowered self-esteem [2,4] occurrence of depression and anxiety [2,5], including becoming suicidal [6].

Treatment of acne includes the use of topical and systemic medications, including isotretinoin [2,7,8,9,10]. Isotretinoin, or 13-cis-retinoic acid, is considered the most effective treatment for acne [7,11,12,13,14,15,16] and is recommended as a first-line drug for the treatment of moderate to severe forms of inflammatory acne [12].

Retinoids, which include isotretinoin, penetrate well into the stratum corneum, with little penetration into the dermis and subcutaneous tissue [17]. They enhance skin immunity, accelerate exfoliation and brighten the skin [18]. They reduce cell proliferation and differentiation, thus lowering the rate of formation of new inflammatory and non-inflammatory changes, including comedones [19]. They also cause epidermal renewal, increase collagen production and also affect immune modulation [19,20,21,22,23,24,25]. In acne, retinoids reduce sebum production and the size of sebaceous glands [26]. Oral isotretinoin is effective in all areas associated with acne pathogenicity such as production of excess sebum, colonization of P. acnes strains and hyperkeratinization of follicles; it also has anti-inflammatory effects by reducing the release of inflammatory mediators [12,27].

However, retinoids, in addition to their positive effects, also show side effects including teratogenic effects [11,21], on the osteoarthritic system and lowering of mood [28]. During retinoid therapy, leukopenia, erythropenia, decreased hematocrit [29], increased transaminase activity [21,30,31], increased cholesterol, LDL-cholesterol and triacylglycerols [32,33] are observed. Retinoids also cause adverse skin symptoms in some patients, such as acne exacerbation, dry skin, erythema, cheilosis or hair loss [21,34,35]. They cause a decrease in stratum corneum hydration and an increase in transepidermal water loss (TEWL) values [26]. Weight gain has also been observed in patients treated with isotretinoin [36,37].

A factor that may reduce the severity of isotretinoin side effects is polyunsaturated fatty acids, a source of which is evening primrose oil (*Oenothera paradoxa*) [38]. This oil stabilizes the keratinization process of the epidermis, exhibits anti-inflammatory effects, plays an important role in skin hydration and prevents excessive water loss from the skin [39]. It has beneficial effects on TEWL, elasticity, firmness, fatigue resistance and skin roughness [40]. The biological effect of evening primrose oil is due to its composition and primarily its content of linoleic acid (LA) and γ-linolenic acid (GLA), which belong to the omega-6 family of acids [38,41]. Both LA and GLA play an essential role in the normal functioning of the skin by, among other things, being part of the ceramides of the epidermal lipid layer, regulating epidermal keratinization, and having strong anti-inflammatory effects [40,42,43,44].

Although several studies have failed to demonstrate the effect of evening primrose oil on weight reduction [45,46], recent studies indicate that it can significantly reduce waist circumference, as well as have beneficial effects on body weight, BMI (Body Mass Index), hip circumference and WHR (Waist Hip Ratio) [47]. Evening primrose oil has also been shown to increase adiponectin levels and insulin sensitivity [48] which may help maintain or reduce body weight in patients treated with isotretinoin.

The purpose of the present study was to evaluate the effects of evening primrose seed oil supplementation in acne vulgaris patients treated with isotretinoin or isotretinoin combined with evening primrose oil on skin hydration status (CORN), transepidermal water loss (TEWL), skin oiliness (sebum) and changes in body weight and BMI.

## 2. Materials and Methods

The randomized trial was conducted in accordance with CONSORT standards (see Appendix A) [49,50]. The study protocol was conducted in accordance with the guidelines of the Declaration of Helsinki. Approval for the study was granted by the Bioethics Committee of the Poznan University of Medical Sciences (ref. 268/15). All participants were informed about the purpose and conduct of the study. They were also informed about the voluntariness of participation and the possibility of resigning at any stage of the study without giving reasons. All patients recruited for the study gave their consent to participate in the study.

### 2.1. Patients

The potential study participants were recruited from among patients at the outpatient clinic of the Department of Dermatology at the Poznan University of Medical Sciences. Based on the dermatological evaluation, 50 participants aged 18 to 30 years with diagnosed acne vulgaris of moderate to severe severity were enrolled in the study. Participants were randomized to two arms: 25 to the IOW study group and 25 to control group I. The groups were homogeneous in terms of age (*p* = 0.116). The mean age of patients in the IOW group was 22.5 ± 1.92 years and 21.6 ± 2.14 years in the control group.

After consenting to participate in the study, a dermatological evaluation was performed among the patients at the inclusion visit to check their compliance with the protocol requirements.

Inclusion criteria included a diagnosis of moderate to severe acne vulgaris and/or no improvement after previous treatment. Exclusion criteria included patients with liver failure, abnormal lipid profile (high total cholesterol and LDL-cholesterol), metabolic diseases, taking statins, high vitamin A levels or hypersensitivity to vitamin A. Pregnant or breastfeeding women were also excluded from the study. The selection of participants in the study was described in the work of Kazmierska et al. [51].

### 2.2. Study Design

Patients participating in the study were randomly allocated to either arm I (isotretinoin treatment) or IOW (isotretinoin and evening primrose oil). The allocation ratio was 1:1. A randomization with a simple procedure based on computerized random numbers was used.

In both arms, the dermatologist individually selected the dose of isotretinoin for each patient ranging from 10 to 40 mg of isotretinoin/day. The dose of isotretinoin taken by patients ranged from 0.5–1 mg/kg body weight/day, resulting in a cumulative dose of 120 to 150 mg isotretinoin/kg body weight over the entire intervention period. Treatment was usually started with low doses of 0.5 mg isotretinoin/kg bw/day. In the IOW arm, patients additionally took encapsulated evening primrose oil at 4 capsules/day, 2 capsules (2 × 510 mg) in the morning and 2 capsules (2 × 510 mg) in the evening. This dose did not exceed the manufacturer’s recommendations.

The intervention duration was 9 months and ran concurrently in both arms. No side effects or adverse events were reported. All included patients (*n* = 50) completed treatment.

This study presents the results of endpoints such as levels of skin hydration (CORN), transepidermal water loss (TEWL), skin oiliness (sebum), and body weight and BMI before and after the intervention.

The study used evening primrose oil in soft capsules, which was extracted from *Oenothera paradoxa*. One serving of the oil (2 capsules of 42.8 g each from Adamed Consumer Healthcare SA, Pieńków, Poland ) consisted of 1020 mg of evening primrose seed oil (Oenothera paradoxa), including linoleic acid (LA) 694 mg, γ-linolenic acid (GLA) 84.8 mg, gelatin and humectants glycerol and sorbitol.

Study participants were advised not to change their previous dietary habits, physical activity or use any skin treatments (cosmetic and/or dermatological) during the study.

To monitor patients’ adherence to the study protocol, especially regular intake of isotretinoin and oil capsules, regular telephone contact was maintained with them. Once a month, participants returned empty oil capsule packets. All measurements and their analysis were carried out at the Poznan Uniwersity of Medical Sciences before and after each intervention period. All outcomes were assessed using identical methods in both arms.

Randomization was performed using computer software (Excel, Microsoft Corp, Redmond, WA, USA). The randomization list was generated by an independent researcher, and the order in which patients were assigned to the I or IOW arms was concealed until the start of the study. Neither the patients nor the study group knew the participant’s group assignment in advance.

An allocation ratio of 1:1 and a type I error probability of α = 0.05 were used to calculate the minimum sample size. The sample size was calculated using Statistica StatSoft 13.3 data analysis software based on the observed changes in total cholesterol levels of 33.2 ± 20.3 mg/dL before isotretinoin treatment and 153.3 ± 20.3 mg/dL after isotretinoin treatment (Akmaz et al. [52]). The calculated minimum sample size was 23 subjects and the power of the test was 0.9072.

### 2.3. Experimental Procedure

#### 2.3.1. Assessment of Skin Condition Parameters

Measurements and assessments of skin condition parameters were performed at baseline and after 9 months of intervention. Non-invasive methods such as a Tevameter (TM 210 ELECTRONIC) for assessing transepidermal water loss, a Sebumeter (SM 810 ELECTRONIC, Koln, Germany) for assessing sebum content and a Corneometer (CM 825 ELECTRONIC, Koln, Germany) for assessing transepidermal skin hydration were used to evaluate the biophysical properties of facial skin [53]. All skin measurements were taken in a sitting position, in a room with a temp of 20–25 °C and relative humidity of 40–60%. The skin status was assessed on a cleaned area of the left cheek (an area 4 cm in diameter). The measurements were preceded by a 30-min acclimatization period in the room where the measurements were taken.

##### Assessment of Transepidermal Hydration of the Skin (CORN)

The hydration status of the stratum corneum was calculated using electrical capacitance measurements with a Corneometer (CM 825). The dielectric constant is a function of the level of skin hydration. Pressing the probe against the cheek results in a change in tissue capacitance. The higher the water content of the epidermis, the higher its electrical capacitance [54], resulting in a higher SC value. Transepidermal hydration was assessed in arbitrary units.

##### Assessment of Skin Lubrication (Sebum)

The quantification of skin surface lipids–sebum lipids was carried out with a Sebumeter (SM 810). The device quantifies the light transmittance of a special matte tape. The light transmittance of the tape changes after 0.5 min of skin contact and depends on the sebum content of the skin surface. The transparency of the tape after the test was assessed using a photo-metric system. Sebumeter measurement was performed at a constant force of 9.4 N/cm^2^. Sebum concentration was quantified in ug/cm^2^.

##### Evaluation of Transepidermal Water Loss (TEWL)

An evaporation meter (Tewameter TM 210, Eindhoven, Netherlands ) was used to determine TEWL. The TEWL value depends on the permeability of the stratum corneum, which is the skin’s protection [53]. Differences between the two measurement points are calculated based on Fick’s diffusion law [55].

#### 2.3.2. Weight and BMI Measurements

Body weight and BMI were measured with a Tanita MC-780 analyzer (Poznań, Poland) using the bioelectrical impedance method. The bioelectrical impedance analysis method involves measuring the resistance that individual tissues exhibit to an electrical impulse acting on them. Water, which is the main component of the body, shows the property of conducting electrical impulses well. Fat tissue, on the other hand, shows greater resistance to the impulse.

Each participant in the study underwent body composition analysis twice-before and after 9 months of intervention. The examinations were performed identically in both arms I and IOW, according to the current methodology, at the same time of day, a minimum of 3 h after the last meal and physical activity. Measurements were taken barefoot, in a standing position, with thighs apart and arms not attached to the body, with hands facing down.

### 2.4. Statistical Methods

The results obtained are presented as median (Me) and quartile deviation (Q). The statistical significance level was set at *p* < 0.05. The Shapiro–Wilk test was used to assess the conformity of quantitative variables to a normal distribution. The assessment of differences before and after treatment was tested with the Wilcoxon–McNemar test, and the assessment of differences between the study groups was checked with the Mann–Whitney U te-stem. All analyses were carried out using the statistical calculation program STATISTICA 13.3 (TIBCO Software Inc., Palo Alto, CA, USA) and PS IMAGO PRO 7 (IBM SPSS Statistics 27) Armonk, NY, USA.

## 3. Results

Recruitment ran from March 2015 to March 2020, and the intervention period from April 2015 to March 2020. Of the 57 subjects enrolled in the study, 7 did not meet the established criteria and were excluded from the study. The remaining 50 participants were randomly assigned to the I arm (*n* = 25) and the IOW arm (*n* = 25). All patients who started the study received the assigned intervention. There were no side effects, and all patients completed the study. The flow chart of the participants is shown in Figure 1.

Both study groups were homogeneous in terms of gender (*p* = 0.7650). The number of patients with moderate and severe acne before the study did not differ significantly between the groups (*p* = 1.000). At the end of the study, the vast majority of patients in both groups showed resolution of their acne lesions, with only 8% in group I and 4% in the IOW group showing a slight aggravation of acne. The difference between the groups was not significant (*p* = 0.5520). The results are shown in Table 1.

Body weight, body height and BMI (Body Mass Index) did not differ between participants assigned to the I arm and the IOW arm before the study (*p* < 0.05). There were also no differences between groups for CORN (*p* = 0.367), TEWL (*p* = 0.786) and sebum (*p* = 705). The initial characteristics of the study population are shown in Table 2.

### 3.1. Evaluation of CORN, TEWL and Sebum Parameters before and after the Intervention

The isotretinoin-treated group showed statistically significant reductions in CORN (*p* = 0.015), TEWL (*p* = 0.004) and sebum (*p* < 0.001) levels after 9 months. In the isotretinoin-treated and evening primrose oil-supplemented group, after 9 months of treatment, TEWL and sebum levels also decreased significantly (*p* < 0.05). However, CORN levels increased significantly from 42.0 ± 9.70 to 50.9 ± 10.4 (*p* = 0.017) in contrast to the isotretinoin-treated group, where a reduction was observed. Between group I and IOW, only the difference for changes obtained for the parameter determining CORN before and after the study was significant (*p* = 0.002). These results are shown in Table 3 and illustrated in Figure 2.

### 3.2. Assessment of Anthropometric Parameters before and after Intervention

In the isotretinoin-treated group, a significant decrease in body weight (*p* < 0.001) and BMI (*p* < 0.001) was observed after 9 months. Similar results were observed in the group treated with isotretinoin combined with evening primrose oil. Here too, a decrease in body weight (*p* < 0.001) and BMI (*p* < 0.001) was found after the intervention. However, there were no significant differences between the groups regarding weight change and BMI (*p* > 0.05). These results are summarized in Table 4 and visualized in Figure 3.

## 4. Discussion

Isotretinoin is effective in the treatment of acne affecting all its principal etiologic factors [19,56,57]. It demonstrates efficacy in the treatment of moderate, severe and recurrent acne, leading to reliable skin improvement [7,11,12,13,14,15,58]. However, there are a number of adverse side effects that limit the use of isotretinoin, including teratogenic effects [11], inducing liver disorders and dysfunction [21,30,59], changes in the lipid profile and especially an increase in total cholesterol, LDL fraction cholesterol, very low-density lipoprotein (VLDL), plasma triglycerides and a decrease in HDL fraction cholesterol [32,33,52,60]. The adverse effects of isotretinoin also affect the skin and mucous membranes. These include dryness of the skin (xerosis) and mucous membranes, dryness in the mouth (xerostomia) and nose, erythema, inflammation, irritation, dryness of the red lips, increased skin sensitivity and conjunctivitis [61,62,63]. Buccal mucositis or dry lips are the most common dose-dependent adverse reactions associated with isotretinoin use, occurring in approximately 90% of patients. Serious skin reactions have also been reported, including Stevens–Johnson syndrome and toxic epidermal necrolysis, warranting immediate discontinuation of isotretinoin [34]. Aksac et al. [64] found a decrease in skin hydration under isotretinoin use and Kmieć et al. [26] found an increase in transepidermal water loss (TEWL) values in addition to a decrease in stratum corneum hydration.

In our study, in a group of patients with acne, treatment with isotretinoin (I) also resulted in a statistically significant decrease in skin hydration (CORN), while supplementation with evening primrose oil resulted in a significant increase in skin hydration in patients treated with isotretinoin, which significantly ameliorated the adverse effects of the drug such as dryness, lip cracking and peeling skin. Evening primrose oil as a dietary supplement is considered safe [65,66]. Although more than 200 drugs interact with evening primrose oil, they show minor interactions [67,68] mainly concerning headache and gastrointestinal side effects [69]. The most serious interactions involve the combination of evening primrose oil with anticonvulsants [70] and drugs used to lower blood pressure [71].

Improvements in skin condition after administration of vegetable oils were also observed by other authors. A study by Bogdan et al. [72] demonstrated the effectiveness of pomegranate seed oil cream (*Punica granatum *L.) in preventing or improving skin lesions associated with stretch marks. Sunflower oil [73,74] and almond oil [73] applied to the skin improved stratum corneum hydration and chia seed oil (*Salvia hispanica *L.) applied topically increased hydration of skin with pruritus [75]. Immunofluorescence (IF) and Western blot analysis of sea buckthorn seed oil (*Hippophae rhamnoides* L.) showed improved skin hydration by increasing the expression of AQP3 and HAS2 in human epidermal keratinocytes [76]. The efficacy of evening primrose oil for skin improvement has been documented by preclinical and clinical studies [77]. Krysiak et al. [78] showed a marked increase in hydration in both dry and normal skin after application of evening primrose oil patches in patients with atopic dermatitis (AD). In adult patients with AD, skin hydration also improved slightly after application of evening primrose oil [79], also in patch form [80]. Orally administered evening primrose oil (*Oenothera paradoxa*) (6 capsules of 500 mg each) improved skin moisture in healthy adults [40].

Assessment of transepidermal water loss (TEWL) is an objective index used to assess skin barrier function [81]. TEWL illustrates the amount of water lost from the body’s interior by diffusion through the epidermis [82,83]. Dysfunction of the skin barrier caused by skin diseases such as atopic dermatitis, contact dermatitis, ichthyosis and psoriasis are associated with higher TEWL values [83]. In contrast, a stronger skin barrier is associated with reduced TEWL [84]. The significant decrease in TEWL observed in our study, both in the group of patients treated with isotretinoin and isotretinoin combined with evening primrose oil supplementation, is indicative of a strengthening of the skin barrier in both patient groups under the influence of isotretinoin treatment. In our study, there were no significant differences between groups in the reduction of TEWL levels.

A reduction in mean TEWL values after 6 weeks of isotretinoin treatment for acne and rosacea was also observed by Kim et al. [85]. On the other hand, Kmieć et al. [26] observed an increase in TEWL values in acne patients receiving isotretinoin therapy, and Colgecen et al., observed no difference in TEWL values after 3 months of isotretinoin treatment [86]. Orally administered evening primrose oil improved TEWL in healthy adults [40].

Excess sebum is the most important factor in the pathophysiology of acne vulgaris [87,88] and is responsible for the formation of acne lesions of a comedonal and inflammatory nature [89]. The main aim of the drugs used to treat acne, including isotretinoin, is to reduce sebum secretion [87]. This is supported by a study by Colgecen et al. [86], in which it was observed that treatment with systemic isotretinoin resulted in a reduction in sebum secretion, and thus improved the biophysical properties of the skin of patients with acne vulgaris. The results of measurements of skin biophysical indices before and after isotretinoin treatment by Kmieć et al. [26] also showed a reduction in the severity of sebum secretion after the intervention. In our study, a reduction in sebum levels was observed in both group I and IOW after 9 months of isotretinoin treatment, and the difference between groups was not statistically significant (*p* > 0.05). Therefore, it can be assumed that the reduction in sebum levels was a result of isotretinoin and that evening primrose oil had no significant effect on the reduction in sebum levels in the intervention group (IOW).

An interesting observation is the significant reduction in body weight and BMI in both study groups after the 9-month intervention (*p* < 0.001). However, our study showed no significant differences between groups in weight loss and BMI. These results differ from those obtained in a study by Cetinozman et al. [36] in women with severe juvenile acne, where weight gain was observed following isotretinoin treatment. In a study by Aydin et al., isotretinoin treatment also resulted in a significant increase in body weight and BMI [37].

The lack of significant differences in weight loss and BMI between the group treated with isotretinoin or isotretinoin combined with evening primrose oil supplementation suggest that the use of evening primrose oil was not the cause of the decrease in these parameters. This is interesting given that evening primrose oil, particularly the γ-linolenic acid (GLA) present in it, by stimulating brown adipose tissue and raising metabolic rate, may be an effective facilitator of weight loss processes; in addition, the resulting prostaglandins trigger fat burning in brown adipose tissue [44]. GLA, by increasing the activity of carnitine palmitoyltransferase and enhancing peroxisomal β-oxidation, may also enhance β-oxidation processes of free fatty acids in the liver and thus accelerate weight loss [90,91].

The main strength of this study was its 9-month, randomized design. This is one of the first studies in humans to evaluate the efficacy of evening primrose oil on skin condition parameters, body weight and BMI during isotretinoin treatment. A strength of the study was also the homogeneity of baseline results between arms I and IOW in terms of gender, acne symptom severity, CORN, TEWL and sebum levels, body weight and BMI. The main limitation of this study was the small group size. However, the calculated minimum sample size was achieved. We also did not assess other factors that could potentially influence the results, such as the use of ointments and creams, cosmetic and dermatological treatments and physical activity. However, all participants were informed not to change their habits in these areas or their dietary habits during the intervention. In our study, no adjustment was made for multiple testing, which could affect the significance of differences in results obtained during the intervention period.

## 5. Conclusions

Considering the results of our study, it was found that during isotretinoin treatment of patients with acne vulgaris, the use of evening primrose oil significantly increases skin hydration, which may mitigate the side effects of treatment with benefits in improving the biophysical characteristics of the skin. There were no differences between the isotretinoin-treated group and the group treated with isotretinoin and supplemented with evening primrose oil on transepidermal water loss and skin oiliness levels. A reduction in body weight and BMI was observed in both groups, but was not dependent on evening primrose oil supplementation.

## Figures and Tables

**Figure 1 nutrients-14-02980-f001:**
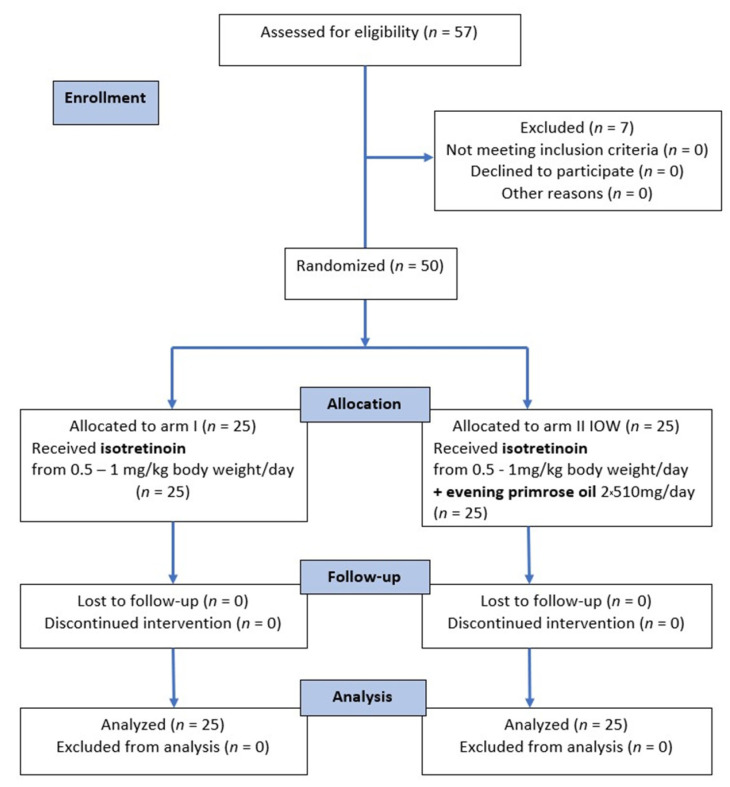
CONSORT 2010 flow diagram [51].

**Figure 2 nutrients-14-02980-f002:**
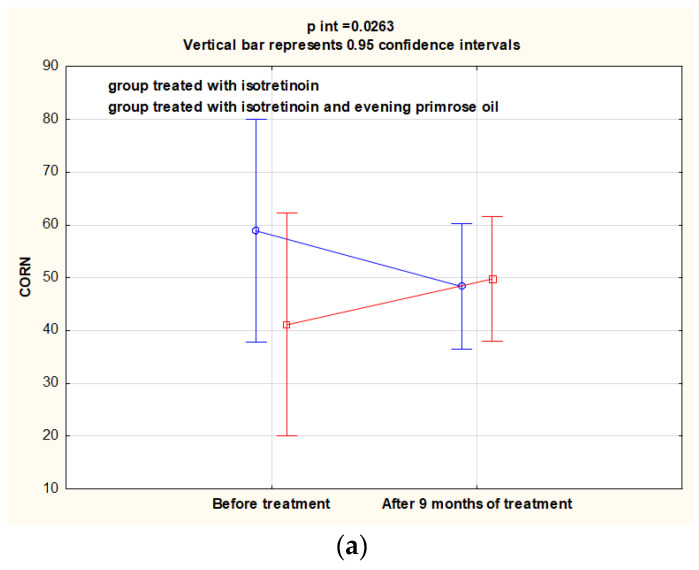
Changes in CORN (**a**), TEWL (**b**), sebum and (**c**) levels in groups I and IOW before and after intervention.

**Figure 3 nutrients-14-02980-f003:**
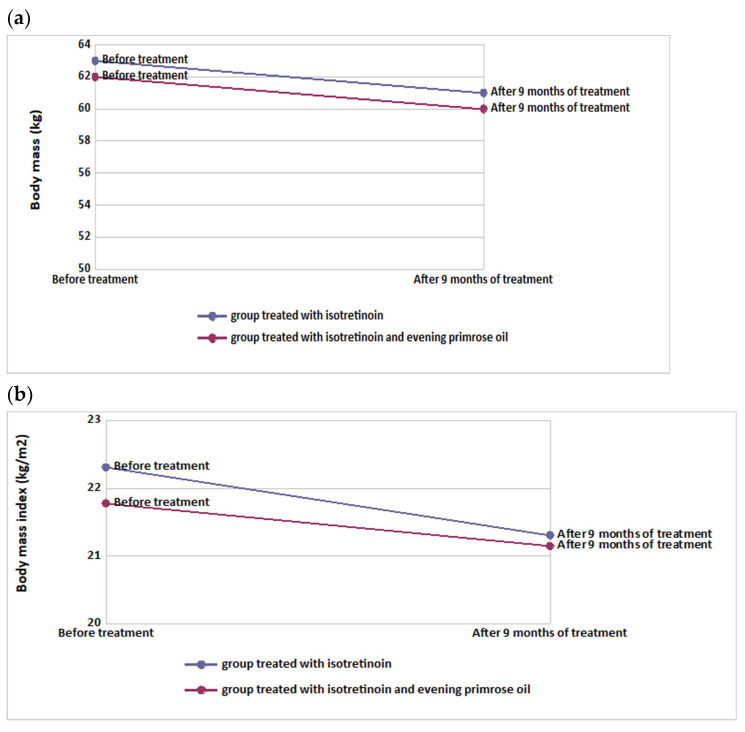
Changes in body weight (**a**) and BMI (**b**) in groups using isotretinoin or isotretinoin with evening primrose oil.

**Table 1 nutrients-14-02980-t001:** Sex of study participants and severity of acne before and after the study.

Analyzed Parameters	Isotretinoin (I) *n* = 25				Isotretinoin with Evening Primrose Oil (IOW) *n* = 25				
Sex of Respondents	Women *n*, %		Men *n*, %		Women *n*, %		Men *n*, %		*p* (between Groups)
	17; 68%		8; 32%		16; 64%		9; 36%		0.7650
Severity of acne before the study	No change *n*; %	Light *n*; %	Medium *n*; %	Hard *n*; %	No change *n*; %	Light *n*; %	Medium *n*; %	Hard *n*; %	
	0; 0%	0; 0%	13; 52%	12; 48%	0; 0%	0; 0%	13; 52%	12; 48%	1.000
Severity of acne after the study	No change *n*; %	Light *n*; %	Medium *n*; %	Hard *n*; %	No change *n*; %	Light *n*; %	Medium *n*; %	Hard *n*; %	
	23; 92%	2; 8%	0; 0%	0; 0%	24; 96%	1; 4%	0; 0%	0; 0%	0.5520

**Table 2 nutrients-14-02980-t002:** CORN, TEWL and sebum levels in group I and IOW before the study started.

Analyzed Parameters	Total Researched *n* = 50				Isotretinoin (I) *n* = 25		Isotretinoin with Evening Primrose Oil (IOW) *n* = 25		*p*
	X ± SD	Me ± Q	Min	Max	X ± SD	Me ± Q	X ± SD	Me ± Q	
Skin Condition Parameters									
CORN arbitrary unit	50.5 ± 32.7	44.1 ± 11.3	6.70	391	58.9 ± 32.4	46.4 ± 10.8	41.1 ± 16.2	42.0 ± 9.70	0.367
TEWL (g/m^2^/h)	20.2 ± 13.7	16.0 ± 6.05	6.00	71.3	22.2 ± 16.5	15.7 ± 9.10	18.1 ± 10.0	17.1 ± 3.20	0.786
Sebum (ug/cm^2^)	92.0 ± 68.5	55.8 ± 30.5	0.00	383	87.8 ± 54.3	65.0 ± 34.5	96.2 ± 64.2	42.0 ± 30.8	0.705
Anthropometric parameters									
Body height (cm)	171 ± 5.79	169 ± 5.50	163	183	170 ± 5.79	169 ± 5.50	171 ± 5.79	169 ± 5.50	0.231
Body weight (kg)	65.0 ± 7.53	62.5 ± 4.50	55.0	86.0	64.6 ± 6.61	63.0 ± 4.00	65.4 ± 8.46	62.0 ± 5.00	0.953
BMI (kg/m^2^)	22.2 ± 1.42	22.0 ± 0.86	20.2	27.8	22.3 ± 1.15	22.3 ± 0.39	22.1 ± 1.67	21.8 ± 1.12	0.367

*n*, number of patients; X, mean; Me, median; Min, minimum value; Max, maximum value; SD, standard deviation; Q, quartile deviation; *p*, statistical significance value of Mann–Whitney U test; CORN, transepidermal hydration; TEWL, transepidermal water loss; sebum, skin lubrication.

**Table 3 nutrients-14-02980-t003:** Changes in CORN, TEWL and sebum levels before and after the intervention.

Analyzed Parameters	Isotretinoin (I) *n* = 25			Wil coxon Test	Isotretinoin with Evening Primrose Oil (IOW) *n* = 25			Wilcoxon Test *p*	U Mann–Whitney Test (Between Groups) *p*		
	Before Treatment	After 9 Months of Treatment	Δ		Before Treatment	After 9 Months of Treatment	Δ				
	Me ± Q	Me ± Q	Me ± Q		Me ± Q	Me ± Q	Me ± Q		Before Treatment	After Treatment	Δ Before and after Treatment
Skin Condition Parameters											
CORN arbitrary unit	46.4 ± 10.8	39.8 ± 12.4	−10.5 ± 38.0	0.015	42.0 ± 9.70	50.9 ± 10.4	8.70 ± 17.6	0.017	0.367	0.190	0.002
TEWL (g/m^2^/h)	15.7 ± 9.10	12.0 ± 6.10	−6.80 ± 14.7	0.004	17.1 ± 3.20	11.5 ± 2.75	−5.80 ± 10.4	0.001	0.786	0.760	0.946
Sebum (ug/cm^2^)	65.0 ± 34.5	14.0 ± 24.0	−59.4 ± 73.8	<0.001	42.0 ± 30.8	5.00 ± 11.5	−79.8 ± 102	<0.001	0.705	0.110	0.554

*n*, number of patients; X, mean; Me, median; Min, minimum value; Max, maximum value; SD, standard deviation; Q, quartile deviation; *p*, statistical significance value; Δ, change; CORN, transepidermal hydration; TEWL, transepidermal water loss; sebum, skin lubrication.

**Table 4 nutrients-14-02980-t004:** Changes in body weight and BMI in groups using isotretinoin or isotretinoin with evening primrose oil.

Analyzed Parameters	Isotretinoin (I) *n* = 25			Wilcoxon Test *p*	Isotretinoin with Evening Primrose oil (IOW) *n* = 25			Wilcoxon Test *p*	U Mann–Whitney Test (Between Grups) *p*
	Before Treatment	After 9 Months of Treatment	Δ		Before Treatment	After 9 Months of Treatment	Δ		
	Me ± Q	Me ± Q	Me ± Q		Me ± Q	Me ± Q	Me ± Q		
Body weight (kg)	63.0 ± 4.00	61.0 ± 4.00	−2.00 ± 0.00	<0.001	62.0 ± 5.00	60.0 ± 5.00	−2.00 ± 0.00	<0.001	0.5250
BMI (kg/m^2^)	22.3 ± 0.39	21.3 ± 0.49	−0.72 ± 0.05	<0.001	21.8 ± 1.12	21.2 ± 1.11	−0.71 ± 0.05	<0.001	0.6686

*n*, abundance; Me, median; Q, quartile deviation; Δ, variance; BMI, Body Mass Index (kg/m^2^).

## Data Availability

Data supporting the results obtained are deposited with the authors for review.

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
