# Peer review of "Effect of Evening Primrose Oil Supplementation on Selected Parameters of Skin Condition in a Group of Patients Treated with Isotretinoin—A Randomized Double-Blind Trial"

_nutrients, 2022, doi:10.3390/nu14142980_

Round 1
Reviewer 1 Report
The study is well structured and developed, however the sample number is not large. Furthermore, there is no objective scale
that assesses the degree of acne severity. It is not expressed if the sample in the two groups is homogeneous by sex: how many males and how many females were present? In line 49 the sentence "Retinoids, which include isotretinoin, penetrate well into the stratum corneum, with little penetration into the dermis and subcutaneous tissue " is unclear, as isotretinoin is an oral retinoid and its action is not limited to the stratum corneum. It is not openly expressed if the primrose oil has contraindications or drug interactions.
Author Response
RESPONSE TO THE REVIEWER'S COMMENTS 1
We would like to thank and express our appreciation to the reviewers and the editorial board for taking the time and effort necessary to improve our work and provide such insightful guidance.
We are very grateful to have been given the opportunity to revise our manuscript, which is entitled " Effect of evening primrose oil supplementation on selected parameters of skin condition in a group of patients treated with isotretinoin - a Randomised Double-Blind Trial."
We have carefully considered your comments. Below we explain how we have revised the paper based on your comments and recommendations.
We have highlighted the corrections and amended sections in red.
Below is a detailed response to the Reviewer:
REVIEWER
The study is well structured and developed, however the sample number is not large.
Response
Thank you very much for your valuable comment. It is very important for us to receive a favourable assessment of the correct design and development of the survey. We agree that the sample is not large, which we highlighted as one of the limitations of the survey. However, given the long period of the survey (9 months) it seems to be sufficient for inference. The minimum sample we calculated was 23 people.
REVIEWER
Furthermore, there is no objective scale that assesses the degree of acne severity.
Response
Thank you very much for your attention. Following the Reviewer's rightful indication, we have included Table 1 in the results section in which we have included information on the severity of acne lesions at the time of eligibility and after 9 months of treatment. This has also been included in the description of the study results. At the same time, we would like to mention that the severity of acne was assessed by the doctor qualifying for the study.
REVIEWER
It is not expressed if the sample in the two groups is homogeneous by sex: how many males and how many females were present?
Response
Thank you very much for your valuable comment. We have attached Table 1, which includes the number of women and men in each study group. There was no statistically significant difference in gender between the study groups. This has also been included in the description of the study results.
REVIEWER
In line 49 the sentence "Retinoids, which include isotretinoin, penetrate well into the stratum corneum, with little penetration into the dermis and subcutaneous tissue " is unclear, as isotretinoin is an oral retinoid and its action is not limited to the stratum corneum.
Response
Thank you very much for pointing out that such an important piece of information was not provided. This has been corrected in the text by adding an additional sentence on the effect of oral isotretinoin in acne and highlighted in red.
Lines 57-60
Oral isotretinoin is effective in all areas associated with acne pathogenicity such as production of excess sebum, colonization of P. acnes strains, and hyperkeratinization of follicles; it also has anti-inflammatory effects by reducing the release of inflammatory mediators.
REVIEWER
It is not openly expressed if the primrose oil has contraindications or drug interactions.
Response
Thank you very much for your valuable comment. We have completed this information and highlighted in red.
Lines 296-301.

Reviewer 2 Report
In this research article the authors present in a very elegant manner all results connected to the evening primrose oil supplementation. Unfortunately, the results of this treatment are not very significant unlike how they are presented in the text.
The authors should extensively revisit the entire discussion and conclusions of the article.
I think is also important to consider in each groups the gravity (moderate or severe) of the acne vulgaris; this could affect all results if the two groups are not homogeneous for severity score.
Lines 230-231. Are you sure about this? All discussion is based on the significant difference in CORN levels between the I and IOW groups
Lines 246-247. The authors write that there are significant differences between groups I and IOW in BMI and body weight. For BMI ok, but for body weight I’m not so sure because to the figure 3a. I suggest double-checking this statistical calculation.
Lines 354-355 “Its supplementation, however, does not have a greater effect on transepidermal water loss and skin oiliness.” The results suggest NO differences between groups I and IOW. I suggest changing the conclusions.
Author Response
RESPONSE TO REVIEWER COMMENTS 2
We would like to thank and express our appreciation to the reviewers and the editorial board for taking the time and effort necessary to improve our work and providing such insightful guidance.
We are very grateful to have been given the opportunity to revise our manuscript, which is entitled " Effect of evening primrose oil supplementation on selected parameters of skin condition in a group of patients treated with isotretinoin - a Randomised Double-Blind Trial."
We have carefully considered your comments. Below we explain how we have revised the paper based on your comments and recommendations.
We have highlighted the corrections and amended sections in red.
Below is a detailed response to the Reviewer:
REVIEWER
In this research article the authors present in a very elegant manner all results connected to the evening primrose oil supplementation. Unfortunately, the results of this treatment are not very significant unlike how they are presented in the text.
Response
Thank you very much for your complimentary opinion on the presentation of the results.
Considering how much of a burden the occurrence and treatment of acne is for patients both physically and psychologically, we addressed this issue. While isotretinoin treatment is effective in reducing acne lesions, it results in other problems such as skin peeling, cracking of the lips, inflammation or irritation of the skin and/or mucous membranes, which causes a lot of aesthetic problems for patients, leading in extreme cases to emotional and social problems. Reducing the adverse effects associated with isotretinoin treatment is important for doctors and patients, hence we took the liberty of presenting the results obtained in the study.
The results we obtained regarding the significant effect of evening primrose seed oil on CORN give great hope for an improvement in skin condition during isotretinoin treatment.
We have attempted to present the results in accordance with the current CONSORT protocol for this type of study, so that they can be used for further comparisons.
REVIEWER
The authors should extensively revisit the entire discussion and conclusions of the article.
Response
Thank you very much for your rightful comment. Following the reviewer's suggestion, we have revised the discussion and conclusions. The changes are marked in red in the text.
REVIEWER
I think is also important to consider in each groups the gravity (moderate or severe) of the acne vulgaris; this could affect all results if the two groups are not homogeneous for severity score.
Response
Thank you very much for your attention. In Table 1 on initial group characteristics, we have completed the information on the number of patients with different acne severity. The groups were homogeneous in terms of severity of acne severity. We have also taken this into account in the text.
REVIEWER
Lines 230-231. Are you sure about this? All discussion is based on the significant difference in CORN levels between the I and IOW groups
Response
Thank you very much for drawing attention to this passage. It is possible that we did not describe clearly enough the changes in CORN, TEWL and sebum parameters after treatment within and between groups. We have therefore amended the description of the results. We hope that this is now presented in a more accessible way.
Lines 245-252
The isotretinoin-treated group showed statistically significant reductions in CORN (p=0.015), TEWL (p=0.004) and sebum (p<0.001) levels after 9 months. In the isotretinoin-treated and evening primrose oil-supplemented group, after 9 months of treatment, TEWL and sebum levels also decreased significantly (p<0.05). However, CORN levels increased significantly from 42.0 ± 9.70 to 50.9 ± 10.4 (p=0.017) in contrast to the isotetinoin-treated group, where a reduction was observed. Between group I and IOW, only the difference for changes obtained for the parameter determining CORN before and after the study was significant (p=0.002).
REVIEWER
Lines 246-247. The authors write that there are significant differences between groups I and IOW in BMI and body weight. For BMI ok, but for body weight I’m not so sure because to the figure 3a. I suggest double-checking this statistical calculation.
Response
Thank you for pointing this out, there was a mistake in table 3 and the description of the results, there was no statistically significant difference in weight reduction and BMI change between groups. The results have been corrected in table 3 and the description, as well as the discussion.
REVIEWER
Lines 354-355 “Its supplementation, however, does not have a greater effect on transepidermal water loss and skin oiliness.” The results suggest NO differences between groups I and IOW. I suggest changing the conclusions.
Response
Thank you very much for Your valuable comment. As suggested by the Reviwer, the conclusions have been corrected.
Lines 377-379
There were no differences between the isotretinoin-treated group and the group treated with isotretinoin and supplemented with evening primrose oil on transepidermal water loss and skin oiliness levels.

Round 2
Reviewer 2 Report
The authors sufficiently improved the manuscript. The results described are not significant, but this type of manuscript is also useful in scientific literature.